# INFERENCE IN PROBABILISTIC GRAPHICAL MODELS BY GRAPH NEURAL NETWORKS

**KiJung Yoon**[1,2]**, Renjie Liao**[3]**, Yuwen Xiong**[3]**, Lisa Zhang**[3]**, Ethan Fetaya**[3]**,**
**Raquel Urtasun**[3]**, Richard Zemel**[3] **& Xaq Pitkow**[1,2]
[1] Department of Neuroscience, Baylor College of Medicine
[2] Department of Electrical and Computer Engineering, Rice University
[3] Department of Computer Science, University of Toronto
{kijung.yoon, xaq}@rice.edu
{rjliao, yuwen, lczhang, ethanf, urtasun, zemel}@cs.toronto.edu

## ABSTRACT

A useful computation when acting in a complex environment is to infer the marginal probabilities or most probable states of task-relevant variables. Probabilistic graphical models can efficiently represent the structure of such complex data, but performing these inferences is generally difficult. Message-passing algorithms, such as belief propagation, are a natural way to disseminate evidence amongst correlated variables while exploiting the graph structure, but these algorithms can struggle when the conditional dependency graphs contain loops. Here we use Graph Neural Networks (GNNs) to learn a message-passing algorithm that solves these inference tasks. We demonstrate the efficacy of this inference approach by training GNNs on an ensemble of graphical models and showing that they substantially outperform belief propagation on loopy graphs. Our message-passing algorithms generalize out of the training set to larger graphs and graphs with different structure.

## 1 INTRODUCTION

Probabilistic graphical models provide a statistical framework for modelling conditional dependencies between random variables, and are widely used to represent complex, real-world phenomena. Given a graphical model for a distribution $p(\mathbf{x})$, one major goal is to compute marginal probability distributions $p_i(x_i)$ of task-relevant variables at each node $i$ of the graph: given a loss function, these distributions determine the optimal estimator. Another major goal is to compute the most probable state, $\mathbf{x}^* = \arg\max_{\mathbf{x}} p(\mathbf{x})$, or MAP (maximum *a posteriori*) inference.

For complex models with loopy graphs, exact inferences of these sorts is often computationally intractable, and therefore generally relies on approximate methods. One important method for computing approximate marginals is the belief propagation (BP) algorithm, which exchanges statistical information among neighboring nodes (Pearl, 1988; Wainwright et al., 2003). This algorithm performs exact inference on tree graphs, but not on graphs with cycles. Furthermore, the basic update steps in belief propagation may not have efficient or even closed-form solutions.

In this work, we introduce end-to-end trainable inference systems based on Graph Neural Networks (GNNs) (Gori et al., 2005; Scarselli et al., 2009; Li et al., 2016), which are recurrent networks that allow complex transformations between nodes. We show how this network architecture is well-suited to message-passing inference algorithms, and have a flexibility that gives them wide applicability even in cases where closed-form algorithms are unavailable. These GNNs have vector-valued nodes that can encode probabilistic information about variables in the graphical model. The GNN nodes send and receive messages about those probabilities, and these messages are determined by canonical learned nonlinear transformations of the information sources and the statistical interactions between them. The dynamics of the GNN reflects the flow of probabilistic information throughout the graphical model, and when the model reaches equilibrium, a nonlinear decoder can extract approximate marginal probabilities or states from each node.

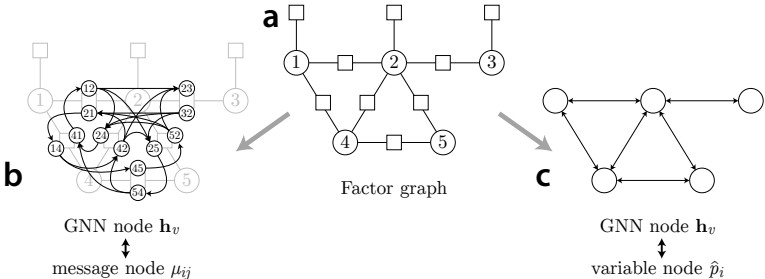

Figure 1: Two mappings between probabilistic graphical model and graph neural network. (a): example graphical model. (b): mapping belief propagation messages $\mu_{ij}$ to GNN nodes $\mathbf{h}_v$. Since different messages flow in each direction, there are two messages per pairwise factor. Each GNN message node is connected to other message nodes that share a variable. (c): mapping variable nodes $i \in \mathcal{V}$ onto GNN nodes $\mathbf{h}_v$. Each GNN node is connected to others that share a factor in the graphical model.

|  | structured | random |
|---|---|---|
| $n = 9$ | I | III |
| $n = 16$ | II | IV |

Table 1. Experimental design: after training on structured graphs with $n = 9$ nodes, we evaluated performance on four classes of graphical models, I-IV, with different sizes ($n = 9$ and $n = 16$) and graph topologies (structured and random) as indicated in the table.

To demonstrate the value of these GNNs for inference in probabilistic graphical models, we create an ensemble of graphical models, train our networks to perform marginal or MAP inference, and test how well these inferences generalize beyond the training set of graphs. Our results compare quite favorably to belief propagation on loopy graphs. See Appendix A for related work.

## 2 GRAPH NEURAL NETWORKS FOR INFERENCE IN GRAPHICAL MODELS

Background materials on probabilistic graphical models, binary Markov random fields, and Graph Neural Networks (GNNs) are available in the Appendix B, C & D. In this section, we present two mappings between graphical models and the GNN (Figure 1). Our experiments show that both perform similarly, and much better than belief propagation.

The first mapping conforms most closely to the structure of conventional belief propagation, by using a graph for the GNN that reflects how messages depend on each other (see Eq 6 in Appendix C). Each node $v$ in the GNN corresponds to a message $\mu_{ij}$ between nodes $i$ and $j$ in the graphical model. GNN nodes $v$ and $w$ are connected if their corresponding message nodes are $ij$ and $jk$ (Figure 1b). If they are connected, messages are computed by $\mathbf{m}_w = \mathcal{M}(\sum_{v:\ell j \| \ell \in N_j \setminus k} \mathbf{h}_v, e_w)$. The readout to extract node marginals or MAP states first aggregates all GNN nodes with the same target by summation, and then applies a shared readout function, $\hat{p}_i(x_i) = \mathcal{R}(\sum_{v:ji | j \in N_i} \mathbf{h}_v)$. This representation grows in size with the number of factors in the graphical model.

The second mapping uses GNN nodes to represent variable nodes in the probabilistic graphical model, and does not provide any hidden states to update the factor nodes (Figure 1c). These factors still influence the inference, since each graphical model's singleton and coupling parameters $J_{ij}$, $b_i$, and $b_j$ are passed into the message function on each iteration. However, this avoids spending representational power on properties that may not change due to the invariances of tree-based reparameterization. In this mapping, the readout $\hat{p}_i(x_i)$ is generated directly from the hidden state of the corresponding GNN node $\mathbf{h}_v$.

## 3 EXPERIMENTS

### 3.1 EXPERIMENTAL DESIGN

Our experiments test how well graph neural networks trained on a diverse set of small graph structures perform on inference tasks. In each experiment we test two types of GNNs, one representing variable nodes (node-GNN) and the other representing message nodes (msg-GNN). We examine generalization under four conditions (Table 1): to unseen graphs of the same structure (I, II), and to completely contrasting random graphs (III, IV). These graphs may be the same size (I, III) or larger (II, IV). For each condition, we examine performance in estimating marginal probabilities and the MAP state. See Appendix E for experimental details.

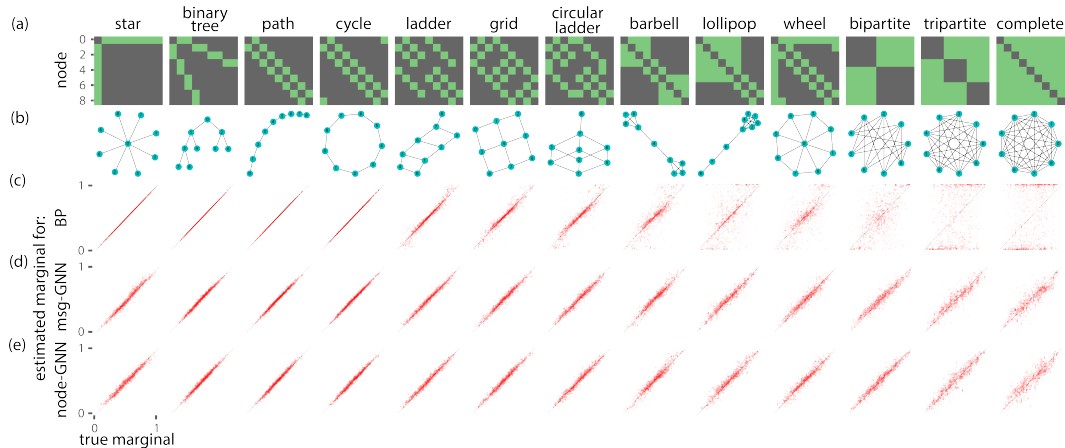

Figure 2: Performance of GNN-based marginal inference on training graphs. (a–b) Example graph structures used in training and testing, shown as adjacency matrices (a) and graphs (b). (c–e) Estimated marginals (vertical axis) are shown against the true marginals for (c) BP, (d) msg-GNN, and (e) node-GNN. Individual red dots reflect the marginals for a single node in one graph. These dots should lie on the diagonal if inference is optimal.

## 3.2 WITHIN-SET GENERALIZATION

To understand the properties of our learned GNN, we evaluate it on different graph datasets than the ones they are trained on. In condition I, test graphs had the same size and structure as training graphs, but the values of singleton and edge potentials differed. We then compared the GNN inferences against the ground truth, as well as against inferences drawn by BP. When tested on acyclic graphs, BP is exact, but our GNNs show impressive accuracy as well (Figures 2c-e). However, as the test graphs became loopier, BP worsened substantially while the GNN inference maintained strong performance (Figures 2c-e).

## 3.3 OUT-OF-SET GENERALIZATION

After training our GNNs on the graph structures in condition I, we froze their parameters, and tested these GNNs on a broader set of graphs.

In condition II (Table 1), we increased the graph size from $n = 9$ to $n = 16$ variables while retaining the graph structures of the training set. In this scenario, scatter plots of estimated versus true marginals show that the GNN still outperforms BP in all of the loopy graphs, except for the case of graphs with a single loop (Figure 3a). We quantify this performance for BP and the GNNs by the average Kullback-Leibler divergence $\langle D_{KL}[p_i(x_i)\|\hat{p}_i(x_i)]\rangle$ across the entire set of test graphs with the small and large number of nodes. We find that performance of BP and both GNNs degrades as the graphs grow. However, except for the msg-GNN tested on nearly fully-connected graphs, the GNNs perform far better than BP, with improvements over an order of magnitude better for graphs with many loops (Figure 3a–b).

To investigate how GNNs generalize to the networks of a different size and structure, we constructed connected random graphs $G_{n,q}$, also known as Erdős-Rényi graphs (Erdős & Rényi, 1959), and systematically changed the connectivity by increasing the edge probability from $q = 0.1$ (sparse) to $0.9$ (dense) for smaller and larger graphs (Conditions III & IV, Figures 3c–d). Our GNNs clearly ourperform BP irrespective of the size and structure of random graphs, although both inference methods show a size- and connectivity-dependent decline in accuracy (Figure 3e). See Appendix F.3 for MAP estimation.

## 4 CONCLUSION

Our experiments demonstrated that Graph Neural Networks provide a flexible method for learning to perform inference in probabilistic graphical models. We showed that the learned representations and nonlinear transformations operating on the edges of the graphical model do generalize to somewhat larger graphs, even to those with different structure. These results support GNNs as an excellent framework for solving difficult inference tasks. Future experiments will consider training and testing on larger and more diverse graphs, as well as on broader classes of graphical models with non-binary variables and more interesting sufficient statistics for nodes and factors.

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

## A    RELATED WORK

Several researchers have used neural networks to implement some form of probabilistic inference. Heess et al. (2013) proposes to train a neural network that learns to map message inputs to message outputs for each message operation needed for Expectation Propagation inference, and Lin et al. (2015) suggests learning CNNs for estimating factor-to-variable messages in a message-passing procedure.

Another related line of work is on inference machines: Ross et al. (2011) trains a series of logistic regressors with hand-crafted features to estimate messages. Wei et al. (2016) applied this idea to pose estimation using convolutional layers and Deng et al. (2016) introduces a sequential inference by recurrent neural networks for the same application domain.

The most similar line of work to the approach we present here is that of GNN-based models. GNNs are essentially an extension of recurrent neural networks that operate on graph-structured inputs (Scarselli et al., 2009; Li et al., 2016). The central idea is to iteratively update hidden states at each GNN node by aggregating incoming messages that are propagated through the graph. Here, expressive neural networks model both message- and node-update functions. Gilmer et al. (2017) recently provide a good review of several GNN variants and unify them into a model called message-passing neural networks. GNNs indeed have a similar structure as message passing algorithms used in probabilistic inference. For this reason, GNNs are powerful architectures for capturing statistical dependencies between variables of interest (Bruna et al., 2014; Duvenaud et al., 2015; Li et al., 2016; Marino et al., 2016; Li et al., 2017; Qi et al., 2017; Kipf & Welling, 2017).

## B  PROBABILISTIC GRAPHICAL MODELS

Probabilistic graphical models simplify a joint probability distribution $p(\mathbf{x})$ over many variables $\mathbf{x}$ by factorizing the distribution according to conditional independence relationships. Factor graphs are one convenient, general representation of structured probability distributions. These are undirected, bipartite graphs whose edges connect variable nodes $i \in \mathcal{V}$ that encode individual variables $x_i$, to factor nodes $\alpha \in \mathcal{F}$ that encode direct statistical interactions $\psi_\alpha(\mathbf{x}_\alpha)$ between groups of variables $\mathbf{x}_\alpha$. (Some of these factors may affect only one variable.) The probability distribution is the normalized product of all factors:

$$p(\mathbf{x}) = \frac{1}{Z} \prod_{\alpha \in \mathcal{F}} \psi_\alpha(\mathbf{x}_\alpha) \tag{1}$$

Here $Z$ is a normalization constant, and $\mathbf{x}_\alpha$ is a vector with components $x_i$ for all variable nodes $i$ connected to the factor node $\alpha$ by an edge $(i, \alpha)$.

Our goal is to compute marginal probabilities $p_i(x_i)$ or MAP states $x_i^*$, for such graphical models. For general graphs, these computations require exponentially large resources, summing (integrating) or maximizing over all possible states except the target node: $p_i(x_i) = \sum_{\mathbf{x} \backslash x_i} p(\mathbf{x})$ or $\mathbf{x}^* = \arg\max_{\mathbf{x}} p(\mathbf{x})$.

Belief propagation operates on these factor graphs by constructing messages $\mu_{i \to \alpha}$ and $\mu_{\alpha \to i}$ that are passed between variable and factor nodes:

$$\mu_{\alpha \to i}(x_i) = \sum_{\mathbf{x}_\alpha \backslash x_i} \psi_\alpha(\mathbf{x}_\alpha) \prod_{j \in N_\alpha \backslash i} \mu_{j \to \alpha}(x_j) \tag{2}$$

$$\mu_{i \to \alpha}(x_i) = \prod_{\beta \in N_i \backslash \alpha} \mu_{\beta \to i}(x_i) \tag{3}$$

where $N_i$ are the neighbors of $i$, *i.e.* factors that involve $x_i$, and $N_\alpha$ are the neighbors of $\alpha$, *i.e.* variables that are directly coupled by $\psi_\alpha(\mathbf{x}_\alpha)$. The recursive, graph-based structure of these message equations leads naturally to the idea that we could describe these messages and their nonlinear updates using a graph neural network in which GNN nodes correspond to messages, as described in the next section.

Interestingly, belief propagation can also be reformulated entirely without messages: BP operations are equivalent to successively reparameterizing the factors over subgraphs of the original graphical model (Wainwright et al., 2003). This suggests that we could construct a different mapping between GNNs and graphical models, where GNN nodes correspond to factor nodes rather than messages. Interestingly, the reparameterization accomplished by BP only adjusts the univariate potentials, since the BP updates lead the multivariate coupling potentials unchanged: after the inference algorithm converges, the estimated marginal joint probability of a factor $\alpha$, namely $B_\alpha(\mathbf{x}_\alpha)$, is given by

$$B_\alpha(\mathbf{x}_\alpha) = \frac{1}{Z} \psi_\alpha(\mathbf{x}_\alpha) \prod_{i \in N\alpha} \mu_{i \to \alpha}(x_i) \tag{4}$$

Observe that all of the messages depend only on one variable at a time, and the only term that depends on more than one variable at a time is the factor itself, $\psi_\alpha(\mathbf{x}_\alpha)$, which is therefore invariant over time. Since BP does not change these interactions, to imitate the action of BP the GNNs need only to represent single variable nodes explicitly, while the nonlinear functions between nodes can account for (and must depend on) their interactions. Our experiments evaluate both of these architectures, with GNNs constructed with latent states that represent either message nodes or single variable nodes.

## C  BINARY MARKOV RANDOM FIELDS

In our experiments, we focus on binary graphical models, with variables $\mathbf{x} \in \{+1, -1\}^{|\mathcal{V}|}$. The probability $p(\mathbf{x})$ is determined by singleton factors $\psi_i(x_i) = e^{b_i x_i}$ biasing individual variables according to the vector $\mathbf{b}$, and pairwise factors $\psi_{ij}(x_i, x_j) = e^{J_{ij} x_i x_j}$ that couple different variables according to the symmetric matrix $J$. Together these factors produce the joint distribution

$$p(\mathbf{x}) = \frac{1}{Z} \exp\left(\mathbf{b} \cdot \mathbf{x} + \mathbf{x} \cdot J \cdot \mathbf{x}\right) \tag{5}$$

In our experiments, each graphical model's parameters $J$ and $\mathbf{b}$ are specified randomly, and are provided as input features for the GNN inference. We allow a variety of graph structures, ranging in complexity from tree graphs to grid graphs to fully connected graphs. The target marginals are $p_i(x_i)$, and MAP states are given by $\mathbf{x}^* = \arg\max_{\mathbf{x}} p(\mathbf{x})$. For our experiments with small graphs, the true values of these targets were computed exactly by exhaustive enumeration of states. Our goal is to construct a recurrent neural network with canonical operations whose dynamics converge to these targets, $p_i(x_i)$ and $\mathbf{x}^*$, in a manner that generalizes immediately to new graphical models.

Belief propagation in these binary graphical models updates messages $\mu_{ij}$ from $i$ to $j$ according to

$$\mu_{ij}(x_j) = \sum_{x_i} e^{J_{ij} x_i x_j + b_i x_i} \prod_{k \in N_i \setminus j} \mu_{ki}(x_i) \tag{6}$$

where $N_i$ is the set of neighboring nodes for $i$. BP provides estimated marginals by $\hat{p}_i(x_i) = \frac{1}{Z} e^{b_i x_i} \prod_{k \in N_i} \mu_{ki}(x_i)$. This message-passing structure motivates one of the two graph neural network architectures we will use below.

## D  GRAPH NEURAL NETWORKS

Graph Neural Networks (Gori et al., 2005; Scarselli et al., 2009; Li et al., 2016) are recurrent networks with vector-valued nodes $\mathbf{h}_v$ whose states are iteratively updated by trainable nonlinear functions that depend on the states of neighbor nodes $\mathbf{h}_w : w \in N_v$ on a specified graph. The form of these functions is canonical, *i.e.* shared by all graph edges, but the function can also depend on properties of each edge. The function is parameterized by a neural network whose weights are shared across all edges. Eventually, the states of the nodes are interpreted by another trainable 'readout' network. Once trained, the entire GNN can be reused on different graphs without alteration, simply by running it on a different graph with different inputs.

Our work builds on a specific type of GNN, the Gated Graph Neural Networks (GG-NNs) (Li et al., 2016), which adds a Gated Recurrent Unit (GRU) (Cho et al., 2014) at each node to integrate incoming information with past states.

Mathematically, each node $i$ in GNN graph $\mathcal{G}$ is associated with a $D$-dimensional hidden state vector $\mathbf{h}_i^{(t)} \in \mathbb{R}^D$ at time step $t$. We initialize this hidden state to all zeros, but our results do not depend on the initial values. On every successive time step, each node sends a message to each of its neighboring nodes. We define the $P$-dimensional vector-valued message $\mathbf{m}_{w \to v}^{t+1} \in \mathbb{R}^P$ from node $w$ to $v$ at time step $t + 1$ by

$$\mathbf{m}_{w \to v}^{t+1} = \mathcal{M}(\mathbf{h}_v^t, \mathbf{h}_w^t, \varepsilon_{wv}) \tag{7}$$

where $\mathcal{M}$ is a message function, here specified by a multilayer perceptron (MLP) with rectified linear units (ReLU). Note that this message function depends the properties $\varepsilon_{wv}$ of each edge $(w \to v)$.

We then aggregate all incoming messages into a single message for the destination node:

$$\mathbf{m}_v^{t+1} = \sum_{w \in N_v} \mathbf{m}_{w \to v}^{t+1} \tag{8}$$

where $N_v$ denotes the neighbors of a node $v$. Finally, every node updates its hidden state based on the current hidden state and the aggregated message:

$$\mathbf{h}_v^{t+1} = \mathcal{U}(\mathbf{h}_v^t, \mathbf{m}_v^{t+1}) \tag{9}$$

where $\mathcal{U}$ is a node update function, in our case specified by another neural network, the gated recurrent unit (GRU), whose parameters are shared across all nodes. The described equations (7, 8, 9) for sending messages and updating node states define a single time step. We evaluate the graph neural network by iterating these equations for a fixed number of time steps $T$ to obtain final state vectors $\mathbf{h}_v^{(T)}$, and then feeding these final node states $\{\mathbf{h}^{(T)}\}$ to a readout function $\mathcal{R}$ given by another MLP with a final sigmoidal nonlinearity $\sigma(x) = 1/(1 + e^{-x})$:

$$\hat{\mathbf{y}} = \sigma\left(\mathcal{R}(\mathbf{h}_{v_i}^{(T)})\right) \tag{10}$$

We train our GNNs using supervised learning to predict target outputs $\mathbf{y}$, using backpropagation through time to minimize the loss function $L(\mathbf{y}, \hat{\mathbf{y}})$.

## E  EXPERIMENTAL DETAILS

Our GNNs are trained on 100 graphical models for each of 13 classic graphs of size $n = 9$ (Figures 2a-b). For each graphical model, we sample coupling strengths from a normal distribution, $J_{ij} = J_{ji} \sim \mathcal{N}(0, 1)$, and sample biases from $b_i \sim \mathcal{N}(0, (\frac{1}{4})^2)$. Our simulated data comprise 1300 training models, 260 validation models, and 130 test models. All of these graphical models are small enough that ground truth marginals and MAP states can be computed exactly by enumeration.

We train GNNs using ADAM (Kingma & Ba, 2014) with a learning rate of $0.001$ until the validation error saturates: we use early stopping with a window size of 20. The GNN nodes' hidden states and messages both have 5 dimensions. In all experiments, messages propagate for $T = 10$ time steps. All the MLPs in the message function $\mathcal{M}$ and readout function $\mathcal{R}$ have two hidden layers with 64 units each, and use ReLU nonlinearities.

We implement our models in Tensorflow (Abadi et al., 2015) and Sonnet.

## F  SUPPLEMENTAL RESULTS

### F.1  OUT-OF-SET GENERALIZATION

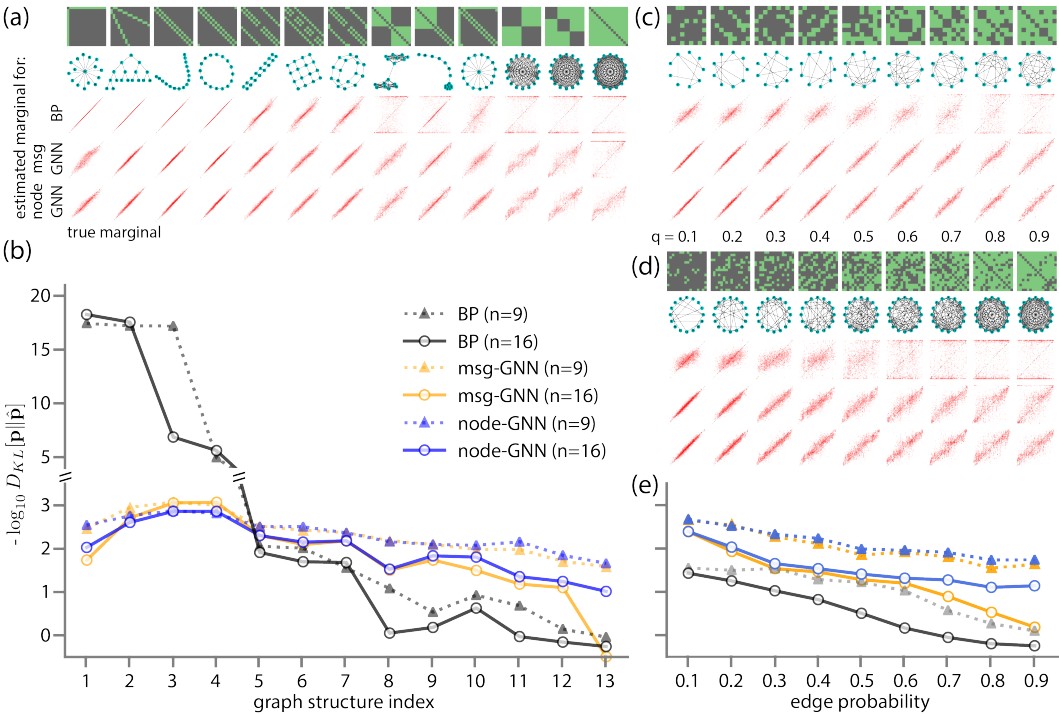

Figure 3: Generalization performance of GNNs to novel graphs. (a) Novel test graphs (larger than the training graphs), and scatter plots of estimated versus true marginals for different inference algorithms, plotted as in Figure 2. (b) Accuracy of marginal inference, measured by negative log KL-divergence in log scale, for graph structures shown above in (a) ($n = 16$, solid lines), and the smaller variants ($n = 9$, dashed lines). Line colors indicate the type of inference method (black: BP, orange: msg-GNN, blue: node-GNN). (c-d) Graphs and scatter plots for random graphs with increasing edge probability $q$, for $n = 9$ nodes (c) and $n = 16$ nodes (d). (e) Generalization performance on random graphs, plotted as in (b).

### F.2  CONVERGENCE OF INFERENCE DYNAMICS

Past work provides some insight into the dynamics and convergence properties of BP (Weiss & Freeman, 2000; Yedidia et al., 2001; Tatikonda & Jordan, 2002). For comparison, we examine how

GNN node hidden states change over time, by collecting the distances between successive node states, $\|\Delta \mathbf{h}_v^t\|_{\ell_2} = \|\mathbf{h}_v^t - \mathbf{h}_v^{t-1}\|_{\ell_2}$. Despite some variability, the mean distance decreases with time independently of graph topologies and size, which suggests reasonable convergence of the GNN inferences (Figure 4), although the rate and final precision of convergence vary depending on graph structures.

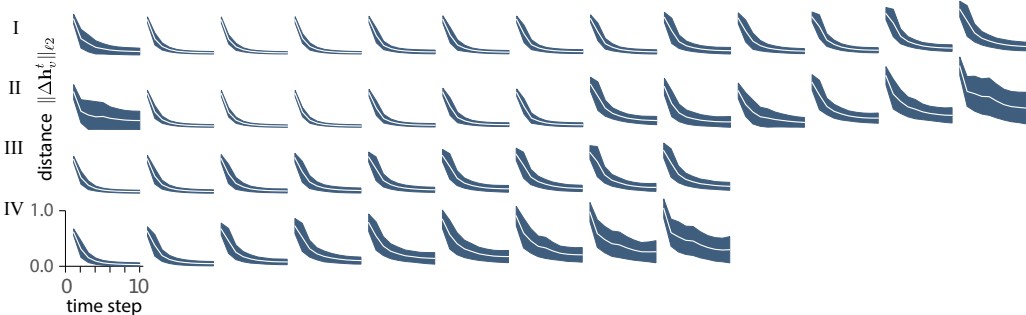

Figure 4: Convergence of GNN inference, measured by the mean (white) and standard deviation (dark blue) of the distances $\|\Delta \mathbf{h}_v^t\|_{\ell_2}$ between successive hidden node states over time. Each row displays the dynamics of GNN on the four experimental conditions I-IV.

### F.3 MAP ESTIMATION

We also apply our GNN framework to the task of MAP estimation, using the same graphical models, but now minimizing the cross entropy loss between a delta function on the true MAP target and sigmoidal outputs of GNNs. As in the marginalization experiments, the node-GNN slightly outperformed the msg-GNN computing the MAP state, and both significantly outperform BP (the max-product variant, sometimes called belief revision (Pearl, 1988)) in these generalization tasks (Figure 5).

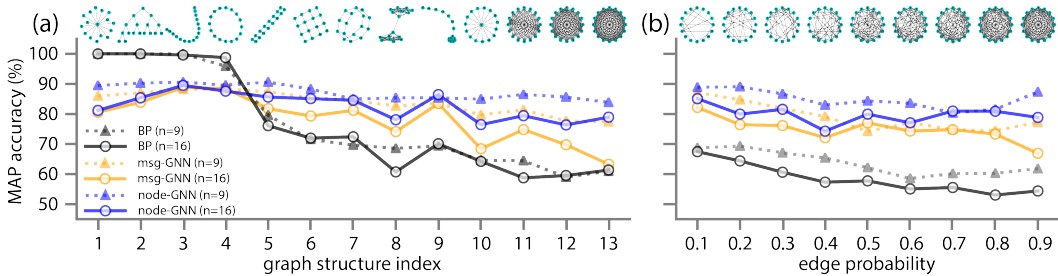

Figure 5: Performance on MAP estimation by GNN inference. (a) Test graphs with $n = 9$ (dashed lines) and $n = 16$ (solid lines) nodes, and probability of correct MAP inference, for BP (black), msg-GNN (orange), and node-GNN (blue). (b) As in (a), but for random graphs of $n = 9$ and $n = 16$ nodes.

