# OpenReview forum: "Inference in probabilistic graphical models by Graph Neural Networks"
_ICLR.cc/2018/Workshop — Accept_

### Official Review · AnonReviewer3 · 2018-03-06
**Ineresting new approach to inference in graphical models**

**Rating:** 7
**Confidence:** 5

**Review:**

This paper proposes using Graph Neural Networks, originally proposed for other applications such as classification on molecular structures, to do inference in pairwise probabilistic graphical models. The setup is fairly straightforward, and really all you need to know to be able to replicate this is the preceding sentence. That said, the experiments are interesting. As expected, anything other than belief propagation does worse than belief propagation on chain and tree graphs; however, on more complex graph structures the graph neural networks seem to be better than BP at predicting node marginals by the metric used (a calibration plot of predicted vs actual marginal probability for each algorithm for each node in a graph for many different potentials). It's also mildly surprising that out of the two versions the paper explored (one with per-factor messages similar to BP and one with simpler per-node messages) the simpler version outperformed the complex one. Specially for max-product inference, for which loopy belief propagation is really bad, this might be useful.

---

### Official Review · AnonReviewer1 · 2018-03-10
**Approach for approximate inference in graphical models using Graph Neural Nets**

**Rating:** 7
**Confidence:** 4

**Review:**

The paper describes an approach to perform inference in graphical models using Graph Neural nets. Specifically, the factor graph in loopy belief propagation is encoded as a graph neural net using 2 encodings, i) encoding variables in the factor graph and ii) encoding messages.

Pros
i) Seems like a nice direction given that gains in approximate inference is highly significant for PGMs
ii) Given the space constraints, has a reasonable evaluation
Cons
i) Complexity of training is not mentioned. For e.g. does BP converge much faster than the proposed GNN method

---

### Official Review · AnonReviewer2 · 2018-03-11
**GNNs for marginal inference**

**Rating:** 7
**Confidence:** 4

**Review:**

The authors propose a method using GNNs for learning to perform marginal inference in graphical models.  The GNN is trained on different graph structures with fixed types of potentials, and the learned models are evaluated on how well the generalize to larger graphs or different potentials versus the belief propagation algorithm for approximate inference.  The results look promising and appear to be novel.

---

### Decision · Program_Chairs · 2018-03-20
**ICLR 2018 Workshop Acceptance Decision**

**Decision:**

Accept

**Comment:**

Congratulations, your paper was accepted to the ICLR workshop.